# Autonomous model protocell division driven by molecular replication

J.W. Taylor [1], S.A. Eghtesadi[2], L.J. Points[1], T. Liu[2] & L. Cronin [1]

The coupling of compartmentalisation with molecular replication is thought to be crucial for the emergence of the first evolvable chemical systems. Minimal artificial replicators have been designed based on molecular recognition, inspired by the template copying of DNA, but none yet have been coupled to compartmentalisation. Here, we present an oil-in-water droplet system comprising an amphiphilic imine dissolved in chloroform that catalyses its own formation by bringing together a hydrophilic and a hydrophobic precursor, which leads to repeated droplet division. We demonstrate that the presence of the amphiphilic replicator, by lowering the interfacial tension between droplets of the reaction mixture and the aqueous phase, causes them to divide. Periodic sampling by a droplet-robot demonstrates that the extent of fission is increased as the reaction progresses, producing more compartments with increased self-replication. This bridges a divide, showing how replication at the molecular level can be used to drive macroscale droplet fission.

---

[1] WestCHEM, School of Chemistry, The University of Glasgow, University Avenue, Glasgow G12 8QQ, UK. [2] Department of Polymer Science, University of Akron, Akron OH 44325, USA. Correspondence and requests for materials should be addressed to L.C. (email: Lee.Cronin@glasgow.ac.uk)

Self-replication is a key property of biological systems at the cellular level, and this drives the process of evolution, but the onset and immediate effects of self-replication in the emergence of life are not understood. One hypothesis is that systems which are able to couple molecular-based template self-replication to cellular objects evolved, overcoming parasitism and problems of dilution, see Fig. 1[1]. To achieve this we propose a strategy that combines the property of molecular replication with that of the formation of a protective bounce of growth and fission, and ultimately objects that could undergo cycles of Darwinian evolution[2, 3]. Such 'model protocells' would allow the idea that life on earth emerged from a minimal self-organised cell-like entity—a protocell, to be tested experimentally[2].

Template self-replication[4] relies on binding of the precursors to the template by molecular recognition, usually combining a hydrogen bonding motif with size, shape and/or charge complementarity. The process may be straightforward and specific, as in the minimal template self-replicator model or more complex, as is the case for the replication of DNA. There are numerous examples of artificial replicators that use molecular recognition to catalyse their own formation by an autocatalytic cycle, but coupling these to other functions that would help the process of evolution is a big challenge[4–6]. However we hypothesised that since a reaction that produces an amphiphile can result in the self-assembly of supramolecular structures such as micelles or vesicles[7], the use of a template self-replication reaction to produce an amphiphile would link self-replication to the formation of such structures. Often the starting materials for these amphiphile formation are hydrophobic (e.g. ethyl caprylate in its hydrolysis to caprylic acid[8]) and therefore the reaction benefits from their solubilisation in micelles, resulting in self-reproduction of the micelles (autopoiesis). Autopoiesis is thought to be a key step in the origin of life due to the necessity of a compartment to prevent molecular information dispersing into the bulk solution, which in turn allows individuals to evolve. A minimal protocell[9, 10] therefore requires compartmentalisation, self-replication (to transfer information from one generation to the next) and metabolism (to utilise material and energy from the environment for growth).

Herein we present our studies on the effects of compartmentalisation on a template-based self-replicator where the template is amphiphilic, and we observe the effects of a self-replication reaction physically on an oil-in-water droplet in a search for emergent system properties. The amphiphilicity of the template is investigated by dynamic light scattering (DLS), and periodic sampling of self-replication reaction mixtures yields an increase in fission of the droplets after placement into an aqueous phase as the reaction progresses.

## Results

**Amphiphilic replicator kinetics.** To investigate the effect of amphiphilicity generation on template-replication processes, we designed an amphiphilic template self-replicator by modification of an existing amidopyridine-carboxylic acid system with a long hydrocarbon chain such that the amidopyridine substrate became the hydrophobic tail, and the carboxylic acid acted as the hydrophilic head[11]. Though only compatible with organic solvents, the reaction could take place in chloroform, with the amphiphilic product stabilizing the chloroform/water interface. By using an imine condensation reaction to join the hydrophilic head and the hydrophobic tail[12], the system could be allowed to reach equilibrium[11]. To investigate our system presented in Fig. 2, the reaction of **1** with **2** in CDCl$_3$ to form amphiphilic imine **3** was followed by 500 MHz $^1$H NMR spectroscopy at 25 °C in dry CDCl$_3$, with (seeded) and without (unseeded) added **3** (15 mol%) at the start of the reaction, see Fig. 3.

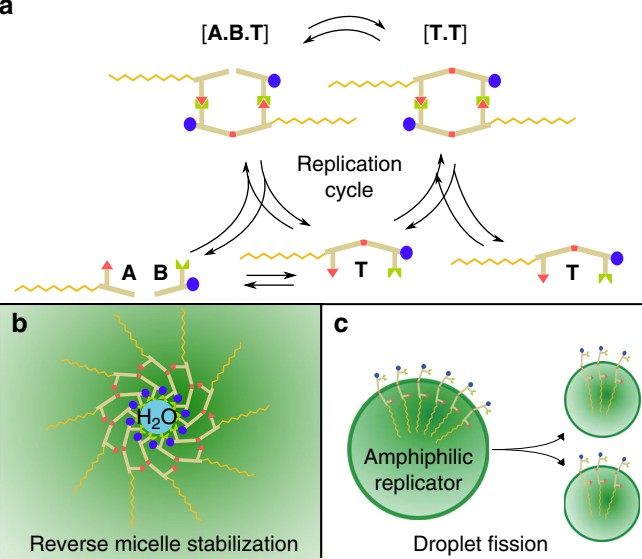

**Fig. 1** Formation of an amphiphile from a template self-replication reaction[20]. **a** Reactants **A** and **B** contain complementary recognition sites and react together via a bimolecular reaction to form template **T**. **T** selectively binds **A** and **B** by the corresponding recognition sites, giving a ternary complex [**ABT**] where the reactive groups of **A** and **B** are perfectly placed to react together, resulting in a rapid pseudounimolecular reaction. The template dimer [**TT**] can then dissociate to give two molecules of **T**, completing the autocatalytic cycle. **b** Reactants **A** and **B** are only weakly amphiphilic, but by bringing together a hydrophobic and a hydrophilic group, amphiphilic **T** is formed. This is demonstrated by its ability to stabilise reverse micelles of water in chloroform. **c** Stabilisation of a droplet chloroform/water interface by amphiphilic **T** allows droplets to undergo fission, as demonstrated by placing microlitre volume droplets from a sample of a reaction mixture periodically using a liquid-handling robot

The unseeded reaction gave a sigmoidal concentration vs. time profile with an induction period characteristic of autocatalytic reactions, as initially the concentration of template was low, and the majority of the template was formed by the bimolecular reaction of **1** and **2**. As [**3**] increased, the rate also increased to a maximum as the reaction increasingly took place via a [**1·2·3**] ternary complex. With the seeded reaction, enough **3** was present at the start of the reaction to ensure the maximum rate was reached at the start of the reaction, with no sigmoidal behaviour. Both curves were fitted to an adaptation of the minimal replicator model[11] using the chemical reaction modelling and curve fitting features of Berkeley Madonna. It can be observed that the modified amphiphilic replicator exhibits a comparable maximum rate (0.41 mM h$^{-1}$) to that of the unmodified replicator (0.47 mM h$^{-1}$) at an initial concentration of 10 mM without seeding with template at the start. Best fit model parameters are compared between the different reactions in Supplementary Table 1. As the association constant for the template dimer (TT) was far higher than the association constant between the template and the substrates (~ 80,000 vs. ~ 34 M$^{-1}$), product inhibition was observed; however, the rapid rates of association and dissociation meant that a small amount of free template (0.1–0.25 mM) was available for binding at any given time, therefore the autocatalytic pathway was a significant contributor to the rate of reaction. The in situ reduction of **3** by sodium triacetoxyborohydride was also attempted, but similarly to previous work in the literature on this type of self-replicating imine system, the precipitation of the amine as its sodium salt prevented its observation by NMR spectroscopy[11].

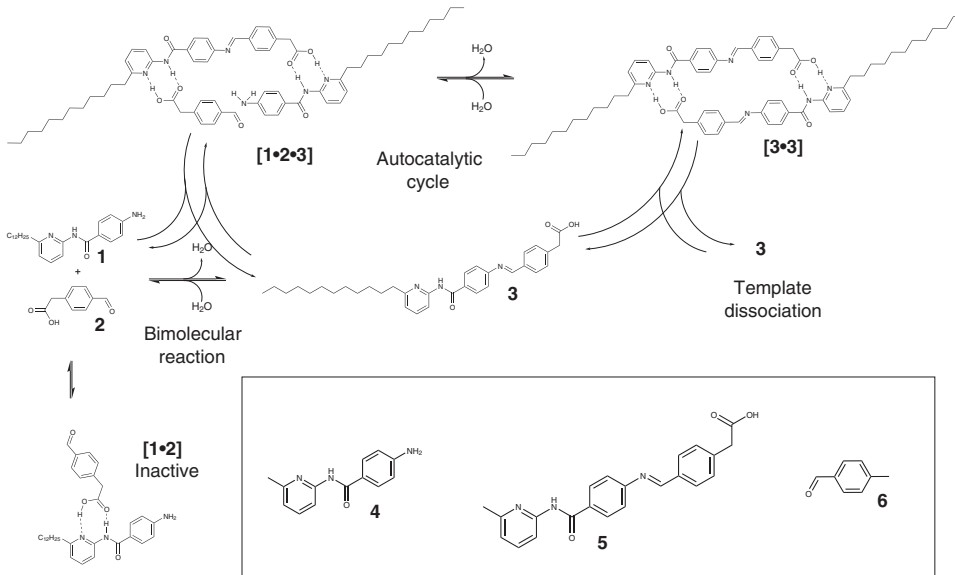

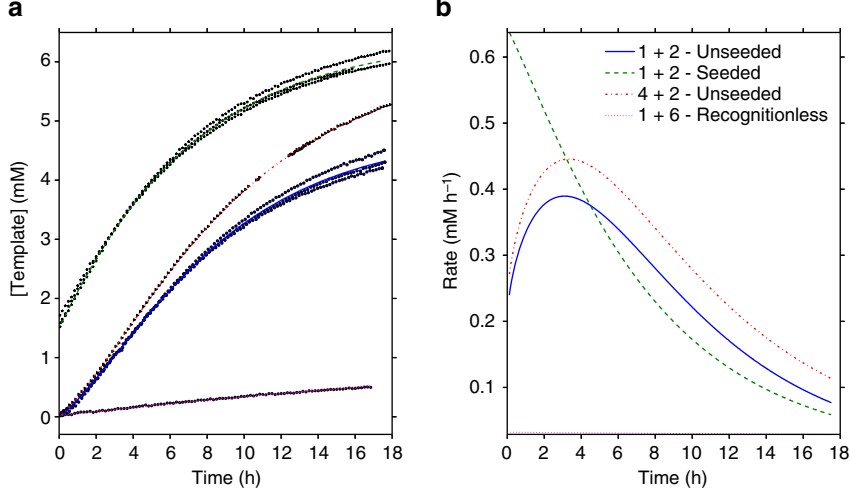

**Fig. 2** Chemical structures. Extending on an existing self-replication motif[11], modified template replicators were designed to form amphiphilic products, and synthesised according to Supplementary Figs. 1 and 2. Hydrophobic amine **1** contains a dodecyl group as a hydrophobic tail, but is only weakly amphiphilic. It reacts with 4-formyl(phenyl) acetic acid **2** in chloroform, which has a carboxylic acid group as its recognition site that doubles as the surfactant head, to form amphiphilic imine **3**. Amine **4** is the original unmodified amine, imine **5** is the unmodified imine, and *p*-tolualdehyde **6** is used as a control substrate as it lacks a recognition site

**Fig. 3** Kinetics of self-replication reaction. **a** Concentration vs. time profiles for the reaction of **1** and **2** in CDCl$_3$, with (*green*) and without (*blue*) seeding with **3**. Also shown are the reactions of unmodified amine **4** with **2** to form **5** (*red*) and of **1** with **6** (*magenta*). All data points obtained over three repeats are plotted along with the model fit, and error analysis for reactions of **1** and **2** is included in Supplementary Figs. 3 and 4. **b** First derivatives of the model fits shown in **a** to give the rate of reaction vs. time for each case

A replicator analogue bearing a branched 3-pentyl-1-undecynyl chain was also investigated for its reaction with **2**, giving a similar kinetic profile (Supplementary Fig. 5) to that of **1** and **2**. From the kinetic data collected with the modified replicator compounds, it can be shown that modification of the pyridine ring in the 6-position does not prevent the replication pathway from operating, and under these conditions, any effect of the longer chain substituents on the kinetics of the replication reaction is minor.

**Dynamic light scattering**. The roles of amphiphilic imine **3** and unmodified imine **5** in encapsulating the trace amount of water in chloroform within the reverse micelles were examined by DLS.

2 mM solutions of each compound in chloroform were mixed with deionised water (0.5% v/v) at pH 12.2 and analysed by time-resolved DLS. Such a small amount of water was used to try to limit the rate of hydrolysis of the imine while still allowing for a small water pool to form the core of reverse micelles. Immediately after sonication, the scattered light intensity of the solution containing compound **3** was 56 times higher than for the solution of **5** which indicates that compound **3** is a more effective amphiphile for stabilizing water in the form of reverse micelles (Supplementary Fig. 6). DLS was also used to monitor the reverse micelle formation during the self-replication reaction of **1** and **2** at 20 mM in dry chloroform. DLS measurements on compound **3** over 36 h showed that the scattered light intensity increased slowly as the reaction was occurring, showing the formation

of reverse micelles with sub-100 nm hydrodynamic radii (Supplementary Fig. 7). Although compound **3** was shown to be able to act as an emulsifier, the stability of the reverse micelles was low, the reverse micelles aggregated quickly and eventually formed a separate phase in each step of the reaction (Supplementary Fig. 6). The low intensity of the scattering was confirmed to be the result of the small amount of water produced by the reaction. To confirm this, volumes of water from 1.25 to 5.00 μl were added to the reaction mixture after completion, resulting in an increase in scattered intensity to a comparable value to that observed in Supplementary Fig. 6 (Supplementary Table 2).

**Automated periodic droplet experiments**. Reactions that form amphiphiles have also been studied in droplets of organic liquids in aqueous solution, for example with the hydrolysis of oleic anhydride[13] or a surfactant imine[14] resulting in self-propelled oil droplets, and the deprotonation of 2-hexyldecanoic acid resulting in droplet division until nanoscale micelles are reached[15], or chemotaxis along a pH gradient[16]. To explore the effects of the formation of **3** in a similar oil droplet system, a hydrophobic dye (sudan blue II) was added to a reaction mixture containing 10 mM of each **1** and **2** in chloroform. Four 4.0 μl droplets of this mixture were placed into dilute sodium hydroxide solution (adjusted to pH 12.2). Initially, the chloroform dispersed on the surface and evaporated, but when samples were taken from the reaction mixture at intervals, clear changes were observed. The droplets would sink below the surface (thus resisting evaporation), divide and move (Supplementary Movie 1). To investigate the changes in droplet behaviour when formed in a consistent manner at constant time intervals, a liquid-handling robot built to study oil droplets (similar to the robot used in previous work from our group on self-propelled oil droplets[17]) was programmed to place four 4.0 μl droplets (in a square pattern) into a petri dish containing the dilute NaOH solution (2.0 ml) at 10 min intervals over the course of the reaction (Fig. 4).

The total area covered by droplets could then be quantified using image tracking software. This showed an increase in droplet area as a result of increased droplet fission (and thus less evaporation) as the reaction progressed, correlating the self-replication reaction with droplet division, (Fig. 5). The total number of droplets also showed an increase (Supplementary Fig. 8), and the average droplet size first increased as droplets sank below the surface and persisted, then decreased as they underwent fission (Supplementary Fig. 9). From this it can be deduced that the amphiphilic replicator **3** is an effective surfactant, and is adsorbed at the chloroform/water interface, lowering the chloroform/water interfacial tension. A higher concentration of **3** enables a higher surface concentration and increases the surface area which can be covered in **3** up to its equilibrium concentration, promoting fission. The robot conducts protocell functions that cannot be fulfilled by the chemistry, in this case protecting the bulk reaction mixture from hydrolysis. The current imine linkage is unstable to aqueous hydrolysis, and solubility issues prevented the use of a reducing agent to convert the product to a stable secondary amine. A variant of the amphiphilic replicator with a non-hydrolysable linkage could potentially operate continuously, requiring the robot only to feed droplets with fresh precursors and organic phase to enable continuous growth and division.

Cross catalysis of the self-replication reaction by unmodified imine **5** was also investigated, to mimic the effects of seeding on the rate of change in droplet division without adding in amphiphile at the start. The same reaction as investigated in Fig. 5 was conducted with 1.5 mM added **5** at the start (Supplementary Fig. 10). This resulted in far greater droplet areas from the start of the reaction, although the extent of the spreading of the droplets resulted in a high variance across three repeats. The mean droplet area over this experiment was 235 vs. 80 mm$^2$ for the second half of the unseeded experiments. The second half of the cross-catalysis experiment was compared to the first half by a single-factor analysis of variance (ANOVA), showing a small but significant difference in average area from 198 to 271 mm$^2$ (ANOVA F factor = 18.6, degrees of freedom = 89, $p = 4.28 \times 10^{-5}$, sample size ($n$) = 90).

## Discussion

By exploring modifications to minimal organic replicators, we were able to demonstrate that an amphiphile can be formed that templates its own formation from two non-amphiphilic building blocks by molecular recognition. Reaction rates were compared between different analogues with different substituents (methyl, $n$-dodecyl and 3-pentyl-1-undecyn-yl) in the 6-position of the amidopyridine ring, and the rate was shown to be fairly insensitive to the bulk of this substituent. From DLS measurements, the emergence of sub-100 nm aggregates was observed over the course of the reaction, and amphiphilic imine **3** was demonstrated to encapsulate water better than the unmodified imine **5**, with over 56 times higher scattering intensity under the same conditions. This did not lead to a rate increase by physical autocatalysis as expected for imine synthesis[18], which can be rationalised as the preference for the starting materials to stay in solution rather than aggregate at the interface or in inverse micelle interiors as does the product. However, the easily observable physicochemical effects of the self-replicating amphiphilic product on the division of oil-in-water droplets demonstrated the coupling of the nanoscale phenomenon of template self-replication to macroscale chemical compartments. The relationship between surfactant concentration and droplet size for micrometre scale droplets has been well investigated previously, and droplet size is known to be highly sensitive to even low concentrations of surfactant[19]. The application of this principle to millimetre scale droplets, and its use to allow template self-

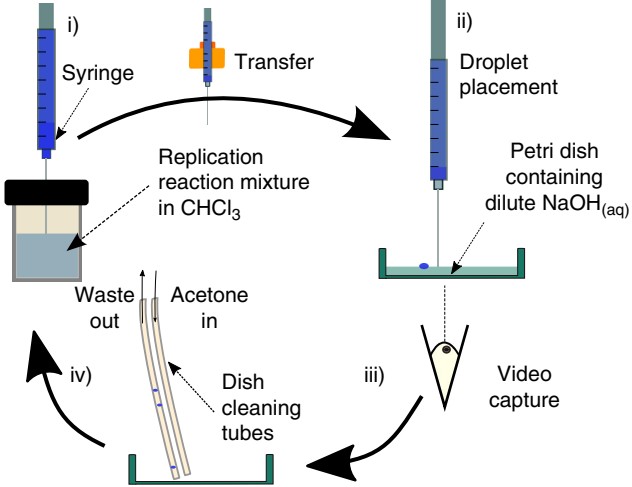

**Fig. 4** Flow diagram showing the periodic droplet experiments. (i) Sampling from reaction mixtures of 10 mM **1** and **2** in chloroform. (ii) 4.0 μl droplets of this chloroform solution of reactants **1** and **2**, amphiphilic product **3**, and the hydrophobic dye are placed into the alkaline aqueous phase (dilute aqueous NaOH, pH 12.2, 2.0 ml) and observed from below. (iii) Video of the resulting droplets is captured for 120 s. (iv) The aqueous phase is aspirated from the dish, and it is cleaned with distilled water and acetone before the next experiment. This was repeated every 10 min for 30 cycles

replication to drive droplet division, forms a protocell model where template self-replication is linked to compartmentalisation, and leads the way for experiments where sustained evolution of populations of replicators is made possible by physicochemical effects determining the survival of a model protocell in the environment. By using the robot for consistent periodic sampling and droplet placement, we were able to demonstrate the link between molecular self-replication and the fate of droplets. Through this iterative process where more functions are developed into the chemical system to end the need for the robotic platform to fulfil them, we envisage increasingly sophisticated protocell models to be achievable with each step. Specifically, a water stable amphiphilic replicator that could form within a droplet would allow the replication reaction to affect the compartment in real time. An even more advanced system where the reaction could take place in aqueous solution could potentially remove the need for an organic phase, with the replicator itself forming bilayer vesicles. These improvements represent worthwhile next steps in the experimental study of linked template self-replication and compartmentalisation.

## Methods

**General information**. The synthesis of compounds **1**–**8** is detailed in Supplementary Methods (NMR spectra in Supplementary Figs. 11–18). All chemicals were purchased from Sigma Aldrich, Fluorochem, TCI organics or Cambridge Isotope Laboratories and used without further purification. Flash chromatography was carried out on a Reveleris X2 flash chromatography system. All NMR spectra were recorded at 25 °C. NMR spectra were measured on a Bruker Avance II 400 MHz, Avance III 400 MHz, Avance III 500 MHz or Avance III 600 MHz spectrometer.

Chemical shifts are reported in ppm relative to the residual solvent peak. Chemical shifts ($\delta$) are given in ppm and coupling constants ($J$) are quoted in hertz (Hz). Resonances are described as s (singlet), d (doublet), t (triplet), q (quartet) and m (multiplet). Electron impact mass spectrometry was carried out on a Jeol MStation JMS-700 high resolution mass spectrometer. Electrospray ionisation mass spectrometry was carried out on a Bruker microTOFq high resolution mass spectrometer (LTQ Orbitrap XL for compounds **7** and **8**). Elemental analysis was measured with an Exeter CE-440 Elemental Analyser.

**Kinetics procedure**. All solutions were made up in dry $CDCl_3$ (redried with further 4 Å molecular sieves), and kept in a dessicator. A screw-cap NMR tube was used to minimise evaporation and exposure to atmospheric moisture. All weighings were carried out with a six decimal place microbalance. A 20 mM solution of the amine was prepared with ~ 1 mM hexamethylbenzene as standard, and the exact concentration recorded. A 20 mM solution of 4-formyl phenylacetic acid was also prepared. For the seeded experiments, the 20 mM amine solution also contained 3.0 mM of **3**. For the reaction of **1** with *p*-tolualdehyde (**6**), 10 mM of 4-bromophenylacetic acid was also added to keep the acidity constant. The amine solution (0.4 ml) was transferred to a screw-cap NMR tube, and spectra of the starting materials obtained. The aldehyde solution (0.4 ml) was then added, and the tube was promptly transferred to a Bruker Avance III 500 MHz NMR spectrometer and the first spectrum acquired. Spectra were acquired every 10 min, for a period of 18 h (unseeded) or 16 h (seeded), totalling 108 or 96 spectra, respectively. The aromatic peaks that did not overlap, along with the imine peak, were integrated using the intser function Topspin 3.5. The hexamethylbenzene peak was used as an internal standard, integrals for each spectrum were individually calibrated to the hexamethylbenzene peak (i.e. no global standard was used). Times were extracted from the Topspin dataset list.

**Automated periodic droplet experiments**. Droplet experiments were carried out on an in-house built liquid-handling robot equipped with 250 µl syringes, an improved version of the chemorobotic platform used in previous work by this lab[17]. The robotic platform consists of a moveable XY carriage equipped with a

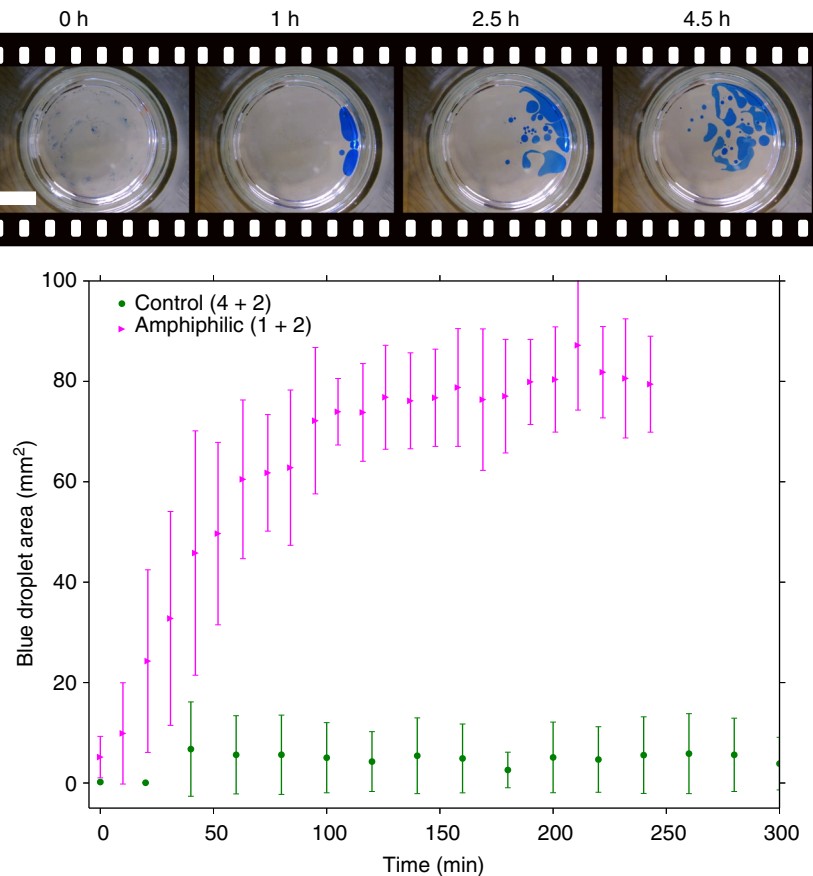

**Fig. 5** Periodic droplet experiment results. Plot of droplet area vs. reaction time, for the self-replicating amphiphile forming reaction starting from 10 mM hydrophobic amine **1** with 10 mM 4-formyl(phenyl) acetic acid **2** (*magenta*) and for the control reaction starting from 10 mM unmodified amine **4** and 10 mM 4-formyl(phenyl) acetic acid **2**. Points are the mean value obtained from three repeats, and the *error bars* show the standard deviation. Above the graph are images taken 120 s after droplet placement from a mixture that had been allowed to react for 0, 1, 2.5 and 4 h respectively (*scale bar* length is 10 mm)

glass syringe that can move in the Z direction, aspirate and dispense droplets. There are also nozzles for dispensing aqueous phase, for pumping out waste and for rinsing the petri dish with acetone. The stage is a transparent glass pane onto which vials and petri dishes can be mounted, and a webcam for visualisation is mounted on another XY carriage below the glass stage. A total of 800 μl aliquots of a 20 mM solution of both the amine under study and 4-formyl(phenyl) acetic acid **2** were mixed in a 1.7 ml vial capped with a PTFE/silicone rubber septum, to which Sudan blue II (3.0 mg) had been added. The robot would then take 30 μl into a glass syringe with a needle through the septum and place four 4.0 μl droplets (in a square pattern) into a 35 mm diameter petri dish containing 2.5 ml of dilute NaOH solution (pH 12.2). A webcam underneath the petri dish recorded video of the droplets for 120 s, then the dish was cleaned with distilled water and acetone, and blown dry with a ducted fan. The droplet placement needle was automatically rinsed in a septum-capped vial containing dry chloroform and 4 Å molecular sieves. This was repeated every 10 min for 30 experiments. From each video, the final frame was taken and the number and total area of droplets was determined using an ImageJ macro. The number of pixels was converted to mm² by dividing the known area of the dish in millimetres by the observed area of the dish from the image in pixels, and multiplying this value by the droplet area in pixels.

**Dynamic light scattering**. DLS measurements were performed using a Brookhaven Instrument spectrometer with solid state laser ($\lambda = 532$ nm) and BI-900AT multichannel digital correlator. Results were later analysed using CONTIN method.

For investigating the roles of amphiphilic imine **3** and unmodified imine **5** in stabilizing traces of water in chloroform (Supplementary Fig. 6), 2.0 mM solutions of each compound were prepared in a glove box while equal amounts of DI water at pH 12.2 (0.5% v/v) were introduced into the solution. The two vials were sealed and mixed thoroughly for a few minutes and then sonicated for 1 min before measurement with time-resolved DLS.

To follow the change in scattering intensity over the course of the self-replication reaction, both compounds were dissolved in anhydrous chloroform separately followed by vortexing, sonication and 30 min of rapid stirring, then both solutions were mixed in a sealed vial, filtered with a 0.2 μm hydrophobic filter (Millipore-Millex-FG) and sealed properly in a glove box. The reaction took place in a dark environment at room temperature with moderate stirring for 36 h. The sealed reaction vial was then opened, and aliquots of water (pH 12.3) up to 1.25, 2.5 and 5 μl were added to 1 ml of the solution, followed by 1 min of sonication. The scatter intensity was then measured (Supplementary Table 2).

**Data availability**. All relevant data are available from the authors upon reasonable request.

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

## Acknowledgements

We gratefully acknowledge financial support from the EPSRC for funding (Grant Nos EP/H024107/1, EP/I033459/1, EP/J00135X/1, EP/J015156/1, EP/K021966/1, EP/K023004/1, EP/K038885/1, EP/L015668/1, EP/L023652/1), BBSRC (Grant No. BB/M011267/1), the EC (projects 610730 EVOPROG, 611640 EVOBLISS), L.C. thanks the Royal Society/Wolfson Foundation for a Merit Award and the ERC for an Advanced Grant (ERC-ADG, 670467 SMART-POM). We are also grateful to the EPSRC UK National Mass Spectrometry Facility at Swansea University for high-resolution mass-spectrometry. T.L. acknowledges support from the National Science Foundation (CHE1607138).

## Author contributions

L.C. conceived the initial concept and together J.W.T. and L.C. developed design approach. J.W.T. performed synthesis and kinetics, S.A.E. performed D.L.S. experiments with input from T.L.; J.W.T. and L.J.P. performed droplet experiments, J.W.T., L.C. and S.A.E. co-wrote the manuscript, with input from the other authors.

## Additional information

**Competing interests:** The authors declare no competing financial interests.

