## [Peer Review File · Nature Communications]

Reviewers' comments:

Reviewer #1 (Remarks to the Author):

It is a challenging manuscript which describes division of an oil droplet induced by increasing concentration of amphiphiles formed through a termolecular self-replication; one molecule is a template and the others are reactants. The division of the oil droplet is driven as a result of a coupling of compartmentation with molecular replication. However, there are following issues to be revised or enriched. I would say this manuscript cannot be published in the present form.

1) It is a smart idea that a condensation between a hydrophobic tail part and a hydrophilic head part to afford the same amphiphile as the template. I wonder the association constant (K_a) between the template and two substrates is considerably smaller than K_a between the template and the product [T T] (the same amphiphile as the template). Otherwise, a turn over the template catalyst becomes very small. In other words, a complex of the template and the product cannot efficiently dissociate into two amphiphilic molecules after the template synthesis is over. Some comments are necessary.

2) The usage of an imine bond for the formation of amphiphiles though dehydro-condensation between aldehyde and amine is important in this work. The authors, however, should cite related papers on the precedent examples of the imine bond formation for a template synthesis of an information molecule and the synthesis of amphiphiles, such as,
A. Terfort, G. Von Kiedrowski, *Angew. Chem. Int. Ed. Engl.* 31, 654-656 (1992),
K. Takakura, T. Toyota, T. Sugawara, *J. Am. Chem. Soc.* 125, 8134 (2003)

3) The kinetic analysis on the formation of the amphiphile with or without the template sounds to be reasonable, but I suspect the formation curve of the amphiphile cannot be designated as a clear sigmoidal one. Generally, a real sigmoidal curve is very hard to obtain in template reactions, as precisely documented by Kiedrowski. The authors should be more cautious about that.

4) In the section of "Dynamic light scattering", the authors claimed that the stabilization of the formed reversed micelle by the amphiphile is not so strong. Hence reverse micelles gradually assembled to form macroscopic oil droplets through the phase-separation. The authors describe that the evaporation of chloroform is negligible since droplets exist in a water phase. My question related to this point is that less volatile solvents, such as dichloroethane, chlorobenzene, cannot be used to avoid the evaporation of the solvent during the measurement.

5) In the section of "Automated periodic droplet experiment", I agree that it is fascinating that an oil droplet (compartmentation) divides into smaller droplets due to the effect of the generated amphiphile through the replication using the template. The size of the divided oil droplet is suspected to be dependent on the concentration of the amphiphile that works as a surfactant. I would be nice to evaluate the concentration of the amphiphile in the divided oil. Such information could be available using a fluorescent probe attached amphiphile.

6) The former part of the manuscript is excellent. However, I am concerned with the discussion in this section. They insist that this event is a coupling of compartmentation with replication of the amphiphile. If this phenomenon is explained from a material side, it could be expressed that the oil droplets divide into pieces by shaking with alkaline water, if the concentration of a surfactant in chloroform becomes higher, which is not extraordinary strange phenomenon.

After all, this work is very challenging as a demonstration of a primitive life model, but the presented data in the last section is still unmaturing. One of the possibilities is that if the authors describe mainly the former part by adding some essential issue of the latter, it may be publishable

in J. Am. Chem.Soc. or in Angewante Chemie.

Reviewer #2 (Remarks to the Author):

The main claim of the paper (paraphrasing the first sentence) is an attempt to demonstrate the coupling of compartmentalization with molecular replication. As the authors further indicate, "There are numerous examples of artificial (molecular) replicators...but coupling these to other functions that would help the process of evolution is a big challenge". The high value of the present manuscript is in taking certain steps towards this ambitious goal. This is of significant interest to a broad audience, will surely influence thinking in the field, and is in principle worthy of publication in the journal.

The crux of the paper is the use of a molecular self replicator which puts together two molecular parts that together generate an amphiphile. The authors adequately demonstrate that the product of this templated adduct formation performs as a functional amphiphile, capable of acting as an emulsifier, leading to the generation (in fact stabilization) of inverted micelles of water within the hydrophobic organic liquid chloroform. The results up to this point already establish a clear connection between a product of templated self-replication and the formation of compartments capable of containing solutes and carrying out further reactions. I would be happy if the authors carried on further studies in this system.

The second part of the paper takes a different route to the main goal - coupling of compartmentalization with molecular replication. The main reason is a wish to demonstrate not only replication-coupled compartmentalization, but also to show coupling to of the replication product to compartment fission. This is highly important, and if adequately done would indeed pave the way to showing a relation of molecular replicators to selection and evolution. The experimental setup selected in this part is following the fate of chloroform droplets introduced into a watery phase. As in the case of inverted micelles, the authors show that the amphiphilic replication product promotes the stabilization of compartmentalized chloroform droplets within the water. A claim is also made that the results demonstrate fission of larger droplets into smaller ones, but this goal remains somewhat elusive, as will be argued in the some of the enumerated points below (marked *). The paper should therefore be revised accordingly to soften some of the claims and to better explain some of the results.

Specific comments (in the order of appearance of the relevant items in the paper)

1. The introduction should be limited to defining the questions at hand, describing what has already been done and what remains to be demonstrated. The more general Fig. 1 is superfluous and should be eliminated. To strengthen this opinion, while Fig 1a is based on published information, Fig. 1b is "blue sky" and its legend contains undemonstrated text such as "allows droplets to persist and divide".
2. Fig. 2 is part of the results and should be placed there.
3. The introduction clause in the introduction "but the role of self-replication in the emergence of life is not known" is overstated and should be modified.
4. The introduction clause "to cellular objects evolved overcoming parasitism and problems of dilution" is irrelevant to the paper's objectives and should be replaced by more appropriate text.
5. The introduction sentence "the formation of a protective boundary to give a system capable of growth, and ultimately objects that could undergo cycles of Darwinian evolution" omits a crucial aspect referred to in the paper – fission.
6. The introduction text about DNA transferring "vast amounts of metastable information" is unclear and irrelevant.
7. The introduction clause "However we hypothesized that a reaction that produces an amphiphile can result in the self-assembly of supramolecular structures such as micelles or vesicles⁸" is not the authors' hypothesis but demonstrated data in reference 8. The hypothesis, as materialized in

the evidence provided in the present paper, does not include compartments formed by assembly of the molecular replicator.

8. In Fig. 2 the legend should include compound 6

9. The compounds first introduced in Fig. 2 are each described by several different terms along the paper. A clear example is calling 3 "dodecyl replicator", "modified elicitor", "amphiphilic imine" and more. The authors should make a special effort to introduce the terminology (with aliases if desired) once and for all here, and use one selected term for each compound in the text that follows.

10. First sentence of the DLS chapter, "The roles of amphiphilic amine (3)" should read "The roles of amphiphilic imine (3)".

11. Chloroform is central to this paper. Along the paper it is referred to also as "organic solvent", "organic phase", "oil", CDCl₃ and more. Better be specific and unified.

12. The sentence immediately preceding the Legend to Fig. 3 is obscure, please clarify.

13. In the DLS chapter, "stabilizing traces of water" is better replaced by a more technically accurate term.

14. Same, "the scattered intensity" should read "the scatter intensity".

15. Same, why call inverse micelles "colloidal particles"?

16. * Regarding the chloroform droplets in water experiment, the general layout is obscure. After some thought the reader may (or may not) see that all it entails is having one vessel contain chloroform, with reactants that go about their chemical reaction in that vessel. That Aliquots are drawn from the vessel, representing different time points of the progress of templated replication, injected into water, and the experiment is about following their fate. Fates include droplet spreading on the water surface, and evaporating, hence not visible as blue entity; getting quickly stabilized and sinking into the water; fusing with other droplets; as is or after fusion undergoing fission.

17. * It is hard to imagine how simply measuring the total blue area can capture the quantitative relationships among all these physicochemical paths. The authors should revise in terms of both the experiments shown and interpretation thereof. In an example, quantitation of any evidence for fission should be shown as a main figure. Otherwise, while it is obvious that the templated surfactant has an effect, it remains undecided whether blue area represents transition from spreading to droplet survival or some other effect. Fission is at a first approximation not expected to augment the blue area.

18. The robotics employed is rather trivial and its description should be in the Methods and the figure – in the supporting material.

19. In the text under legend to Fig. 4, the discussion on equilibrium and away from equilibrium is rather obscure, may be inaccurate, and is not sufficiently rigorous to become relevant to the main points of the paper.

20. In the conclusions, the author relate to freeing the experiment system from robotics. I assume this means that the templating reaction, now in a separate vessel and dropped in by robot, will become an integral part of what happens in the compartment. This should be more clearly stated, and allusion should be made to the exciting possibility that the templated amphiphile will in itself from the compartment (e.g. in the form of a bilayer).

Reviewer #3 (Remarks to the Author):

The authors describe an attempt for fulfilling one of the requirements for a chemical system to be a "minimal cellular system" according to the chemoton theory of Tibor Ganti. In this theory, a minimum model of a chemoton has three self-producing systems which are coupled together stoichiometrically: metabolism cycle, template polycondensation, and membrane formation. On paper, the chemoton theory is plausible. However, an experimental realization of this theoretical view is a big challenge, and if one would succeed, one could claim that a living system (life) could be made from bottom-up. This has not been realized yet. The authors carried out experiments for coupling template copying with compartment multiplication for developing a droplet-based model

of a protocell. More experimental work in this field should be carried out, independent on whether the molecules and conditions used are prebiotic plausible or not, so that new ideas concerning the origin of the first cells, i.e. the origin of life, hopefully emerge. In this sense, the work presented is a valuable contribution.

I recommend considering acceptance of the manuscript once the following points are taken into account (major revision).

1. General: It requires some time to fully understand the details of the system investigated. Therefore, things should be made clearer at different places, including in the abstract. It should be mentioned that the system described is an oil-in-water droplet system and that hydrophobic molecules which are not (or only weakly) amphiphilic are dissolved inside the droplets are converted into amphiphilic molecules via a reaction with a water soluble molecule which takes place through a template mechanism inside the droplets. Is this what the system is?
2. Figure 1: For a, the solvent should be mentioned. For b and c, it should be made clear that the droplets shown contain chloroform inside (okay?). What is the typical size of the droplet? For b it is not clear what the very small dots represent.
3. Title: ... model protocell ... It is a chemical model system of a protocell. Nobody knows how such (hypothetical) protocells looked like.
4. Figure 2, legend. It is not clear whether the entire Figure is taken from ref. 12, or only the templating part. It seems that the work presented is an extension of the work presented in ref. 12. If so, this should be clearly mentioned. Please clarify! It is conceptually important that 1 is not (or only weakly) amphiphilic, while 3 is amphiphilic.
5. Text on p. 5, above Figure 3 is not clear. Is a word missing? ... the reaction could take place in a dispersed organic phase ...
6. Figure 3: The solvents used should be mentioned.
7. Dynamic light scattering: It should be better argued and clearly explained why reverse micelles were used, why not an oil in water systems, a water-in-oil microemulsion.
8. Page 8: a hydrophobic dye (sudan blue II) ...
9. Figure 4a: The blue solution should be explained: reaction systems composed of chloroform droplets in water (okay ?). Figure 4b: Video observation, the same as observation via webcam in Figure 4c? The drawing in Figure 4c is not clear. Why a blue background? Moveable white background? What is the orange part. It is not seen in Figure 4b?
10. Methods: All chemicals purification. The sentence is written two times. Copy paste in the Supporting Information, also two times.

Referee comments in italics; our reply in normal type

Reviewer #1 (Remarks to the Author):

It is a challenging manuscript which describes division of an oil droplet induced by increasing concentration of amphiphiles formed through a termolecular self-replication; one molecule is a template and the others are reactants. The division of the oil droplet is driven as a result of a coupling of compartmentation with molecular replication. However, there are following issues to be revised or enriched. I would say this manuscript cannot be published in the present form.

1) It is a smart idea that a condensation between a hydrophobic tail part and a hydrophilic head part to afford the same amphiphile as the template. I wonder the association constant (K_a) between the template and two substrates is considerably smaller than K_a between the template and the product [T T] (the same amphiphile as the template). Otherwise, a turn over the template catalyst becomes very small. In other words, a complex of the template and the product cannot efficiently dissociate into two amphiphilic molecules after the template synthesis is over. Some comments are necessary.

We had considered the parameters from the model fit in some detail to understand why the system still works with such a large difference in K_a values between A/B /T and TT. Most of the template is present as the [TT] dimer at any given time, so the system does exhibit significant template inhibition, but the rates of association/dissociation are fast enough that this is not entirely rate limiting, and so the formation of product due to the self-replication pathway is still measurable. This difference in K_a values is consistent with Ref. 12 which describes the system ours expands upon.

Looking at the concentrations of species in the model, free [T] is present at just under 0.25 mM by the end of the reaction, and the sigmoidal curve disappears if the self-replication pathway is removed from the model while keeping everything else constant.

We have added the following to the discussion of the kinetic data: .

“As the association constant for the template dimer (TT) was far higher than the association constant between the template and the substrates (80457 M^{-1} vs 34.47 M^{-1}), product inhibition was observed, however the rapid rates of association and dissociation meant that a small amount of free template (0.1-0.25 mM) was available for binding at any given time, therefore the autocatalytic pathway was a significant contributor to the rate of reaction. “

The observed effect on reverse micelles and oil-in-water droplets may have been enhanced by the use of an alkaline aqueous phase, which deprotonates the carboxyl group at the interface, removing its hydrogen bonding capability and increasing its polarity.

*2) The usage of an imine bond for the formation of amphiphiles though dehydro-condensation between aldehyde and amine is important in this work. The authors, however, should cite related papers on the precedent examples of the imine bond formation for a template synthesis of an information molecule and the synthesis of amphiphiles, such as, A. Terfort, G. Von Kiedrowski, *Angew. Chem. Int. Ed. Engl.* 31, 654-656 (1992), K. Takakura, T. Toyota, T. Sugawara, *J. Am. Chem. Soc.* 125, 8134 (2003)*

We thank the reviewer for bringing our attention to the need to cite the precedent for amphiphile formation using imine condensation. We have added a reference to, *J. Am. Chem. Soc.* 125, 8134 (2003). For self-replication, we had cited *Org. Lett.* 10, 4589–4592 (2008) as a relevant example, along with *Eur. J. Org. Chem.* 5, 593–610 (2009) which reviews the development of self-replicating systems, including the one suggested.

3) The kinetic analysis on the formation of the amphiphile with or without the template sounds to be reasonable, but I suspect the formation curve of the amphiphile cannot be designated as a clear sigmoidal one. Generally, a real sigmoidal curve is very hard to obtain in template reactions, as precisely documented by Kiedrowski. The authors should be more cautious about that.

Kiedrowski documents the susceptibility of self-replicating systems to product inhibition with reference to an amidinium/carboxylate salt bridge as a recognition motif, which is far stronger than the amidopyridine/carboxylic acid recognition elements used in this work for which Kiedrowski also reports a sigmoidal curve (*J. Syst. Chem.* 1, 10 (2010)). However, we appreciate that due to this product inhibition, the sigmoidal curve is less visibly clear, though the delayed onset of maximum rate is readily apparent, and the text has been reworded to:

“The unseeded reaction gave a sigmoidal concentration vs time profile with an induction period characteristic of autocatalytic reactions, “

4) In the section of “Dynamic light scattering”, the authors claimed that the stabilization of the formed reversed micelle by the amphiphile is not so strong. Hence reverse micelles gradually assembled to form macroscopic oil droplets through the phase-separation. The authors describe that the evaporation of chloroform is negligible since droplets exist in a water phase. My question related to this point is that less volatile solvents, such as dichloroethane, chlorobenzene, cannot be used to avoid the evaporation of the solvent during the measurement.

It's possible that dichloroethane or dibromoethane would solve any problems of water evaporation. However, phase separation currently leads to chloroform droplets below the surface of the aqueous phase, which also prevents phase separation. We use chloroform here as we have fully investigated the kinetics of the reaction in this solvent and know that it fits the theoretical model. The evaporation of the chloroform is also a feature of the phenomenon observed in the periodic droplet experiments, as without the amphiphile present in the droplet, the surface tension of the aqueous phase prevents the droplet from sinking and results in its evaporation. We decided to continue using chloroform in the DLS experiments in order to preserve the same solvent system across kinetics, DLS, and droplet experiments.

5) In the section of “Automated periodic droplet experiment”, I agree that it is fascinating that an oil droplet (compartmentation) divides into smaller droplets due to the effect of the generated amphiphile through the replication using the template. The size of the divided oil droplet is suspected to be dependent on the concentration of the amphiphile that works as a surfactant. I would be nice

to evaluate the concentration of the amphiphile in the divided oil. Such information could be available using a fluorescent probe attached amphiphile.

We thank the reviewer for their suggestion. In-situ fluorescence measurements of the droplets containing the amphiphile would be fascinating – and indeed we cite a paper where a fluorescent probe attached replicator is used to follow a self-replication reaction. Such an experiment would require new replicator analogues to be designed and synthesized, and their kinetics measured in order to confirm that replication still occurs. This represents a significant new body of work, and is a direction we're considering for our future research into this area.

6) The former part of the manuscript is excellent. However, I am concerned with the discussion in this section. They insist that this event is a coupling of compartmentation with replication of the amphiphile. If this phenomenon is explained from a material side, it could be expressed that the oil droplets divide into pieces by shaking with alkaline water, if the concentration of a surfactant in chloroform becomes higher, which is not extraordinary strange phenomenon.

In the DLS experiments, sonication of the chloroform/water mixture broke up the organic phase into pieces, giving reverse micelles and demonstrating that the replicator also functioned as a surfactant as we had hoped. In the droplet experiments, the mixture was not shaken – and instead changes in interfacial tension drove the division of the droplets. This was not unexpected, indeed self-division of droplets driven by formation of an amphiphile is widely studied in the field of protocells (e.g. *Angew. Chem. Int. Ed.* **49**, 6756–6759 (2010)), but here instead of deprotonation or hydrolysis at the interface, the amphiphile is produced by a template self-replication reaction which links the two important processes, and this is what we set out to demonstrate.

After all, this work is very challenging as a demonstration of a primitive life model, but the presented data in the last section is still unmaturred. One of the possibilities is that if the authors describe mainly the former part by adding some essential issue of the latter, it may be publishable in J. Am. Chem.Soc. or in Angewante Chemie.

Reviewer #2 (Remarks to the Author):

The main claim of the paper (paraphrasing the first sentence) is an attempt to demonstrate the coupling of compartmentalization with molecular replication. As the authors further indicate, “There are numerous examples of artificial (molecular) replicators...but coupling these to other functions that would help the process of evolution is a big challenge”. The high value of the present manuscript is in taking certain steps towards this ambitious goal. This is of significant interest to a broad audience, will surely influence thinking in the field, and is in principle worthy of publication in the journal.

The crux of the paper is the use of a molecular self replicator which puts together two molecular parts that together generate an amphiphile. The authors adequately demonstrate that the product of this templated adduct formation performs as a functional amphiphile, capable of acting as an

emulsifier, leading to the generation (in fact stabilization) of inverted micelles of water within the hydrophobic organic liquid chloroform. The results up to this point already establish a clear connection between a product of templated self-replication and the formation of compartments capable of containing solutes and carrying out further reactions. I would be happy if the authors carried on further studies in this system.

*The second part of the paper takes a different route to the main goal - coupling of compartmentalization with molecular replication. The main reason is a wish to demonstrate not only replication-coupled compartmentalization, but also to show coupling to of the replication product to compartment fission. This is highly important, and if adequately done would indeed pave the way to showing a relation of molecular replicators to selection and evolution. The experimental setup selected in this part is following the fate of chloroform droplets introduced into a watery phase. As in the case of inverted micelles, the authors show that the amphiphilic replication product promotes the stabilization of compartmentalized chloroform droplets within the water. A claim is also made that the results demonstrate fission of larger droplets into smaller ones, but this goal remains somewhat elusive, as will be argued in the some of the enumerated points below (marked *). The paper should therefore be revised accordingly to soften some of the claims and to better explain some of the results.*

Specific comments (in the order of appearance of the relevant items in the paper)

1. The introduction should be limited to defining the questions at hand, describing what has already been done and what remains to be demonstrated. The more general Fig. 1 is superfluous and should be eliminated. To strengthen this opinion, while Fig 1a is based on published information, Fig. 1b is "blue sky" and its legend contains undemonstrated text such as "allows droplets to persist and divide".

We appreciate the reviewer's feedback on Figure 1, and understand the criticism of Figure 1 as too general, but representing the core concept of the work in a graphical way before presenting chemical structures and kinetic data is important to the reader's understanding in our opinion. We have revised Figure 1 to address the criticism of reviewers 2 and 3, and we hope that the reader finds it useful in illustrating the core concept of the paper.

2. Fig. 2 is part of the results and should be placed there.

We have moved Fig. 2 to the results.

3. The introduction clause in the introduction "but the role of self-replication in the emergence of life is not known" is overstated and should be modified.

We agree that this clause does not address the objectives of this work specifically, and have modified it to:

“but the onset and immediate effects of self-replication in the emergence of life are not understood”.

4. *The introduction clause “to cellular objects evolved overcoming parasitism and problems of dilution” is irrelevant to the paper’s objectives and should be replaced by more appropriate text.*

Parasitism may be beyond the objectives of the paper, but in this case we wish to highlight the importance of compartmentalization for the origin of life, especially with reference to competing self-replicating systems. One of the key advantages in compartmentalized systems is the resistance to parasitism, so this frames self-replication and compartmentalization together in the context of protocells.

5. *The introduction sentence “the formation of a protective boundary to give a system capable of growth, and ultimately objects that could undergo cycles of Darwinian evolution” omits a crucial aspect referred to in the paper – fission.*

We agree with this suggestion, and have modified this to:

“the formation of a protective boundary to give a system capable of growth and fission, and ultimately objects that could undergo cycles of Darwinian evolution.”

6. *The introduction text about DNA transferring “vast amounts of metastable information” is unclear and irrelevant.*

We agree that “vast amounts of metastable information” is not essential to our comparison, and have modified this to:

“The process may be straightforward and specific, as in the minimal template self-replicator model or more complex, as is the case for the replication of DNA.”

7. *The introduction clause “However we hypothesized that a reaction that produces an amphiphile can result in the self-assembly of supramolecular structures such as micelles or vesicles⁸” is not the authors’ hypothesis but demonstrated data in reference 8. The hypothesis, as materialized in the evidence provided in the present paper, does not include compartments formed by assembly of the molecular replicator.*

We have revised the text to better reflect the work carried out in this paper:

“However we hypothesised that since a reaction that produces an amphiphile can result in the self-assembly of supramolecular structures such as micelles or vesicles,⁸ the use of a template self-replication reaction to produce an amphiphile would link self-replication to the formation of such structures. Often the starting materials for these amphiphile formation are hydrophobic... “

8. *In Fig. 2 the legend should include compound 6*

We have added compound 6 to the legend of Fig. 2.

9. The compounds first introduced in Fig. 2 are each described by several different terms along the paper. A clear example is calling 3 “dodecyl replicator”, “modified elicitor”, “amphiphilic imine” and more. The authors should make a special effort to introduce the terminology (with aliases if desired) once and for all here, and use one selected term for each compound in the text that follows.

We agree that this would clarify the paper significantly, and have standardized the terminology to:

1 – hydrophobic amine

2 – 4-formyl(phenyl)acetic acid

3 - amphiphilic imine

4 – unmodified amine

5 - unmodified imine

10. First sentence of the DLS chapter, “The roles of amphiphilic amine (3)” should read “The roles of amphiphilic imine (3)”.

We apologize for the error, and have corrected this in the text.

11. Chloroform is central to this paper. Along the paper it is referred to also as “organic solvent”, “organic phase”, “oil”, CDCl₃ and more. Better be specific and unified.

We have changed these references to “chloroform” when appropriate.

12. The sentence immediately preceding the Legend to Fig. 3 is obscure, please clarify.

We have removed this sentence referring to the reduction of self-replicating imines in-situ, since this is also explained on page 7, and we hope that this makes sense in context.

13. In the DLS chapter, “stabilizing traces of water” is better replaced by a more technically accurate term.

We have consulted with our co-authors who performed the DLS study and replaced this sentence with:

“The roles of amphiphilic imine **3** and unmodified imine **5** on encapsulating the trace amount of water in chloroform within the reverse micelles were examined by the Dynamic Light Scattering (DLS).”

14. Same, “the scattered intensity” should read “the scatter intensity”.

We have corrected this in the text.

15. Same, why call inverse micelles “colloidal particles”?

We have standardized the terminology to use “reverse micelles” throughout.

*16. * Regarding the chloroform droplets in water experiment, the general layout is obscure. After some thought the reader may (or may not”) see that all it entails is having one vessel contain chloroform, with reactants that go about their chemical reaction in that vessel. That Aliquots are drawn from the vessel, representing different time points of the progress of templated replication, injected into water, and the experiment is about following their fate. Fates include droplet spreading on the water surface, and evaporating, hence not visible as blue entity; getting quickly stabilized and sinking into the water; fusing with other droplets; as is or after fusion undergoing fission.*

We appreciate the feedback on this section, and have taken the opportunity to replace figure 4 with a flow diagram which we hope explains the experiment more clearly.

*17. * It is hard to imagine how simply measuring the total blue area can capture the quantitative relationships among all these physicochemical paths. The authors should revise in terms of both the experiments shown and interpretation thereof. In an example, quantitation of any evidence for fission should be shown as a main figure. Otherwise, while it is obvious that the templated surfactant has an effect, it remains undecided whether blue area represents transition from spreading to droplet survival or some other effect. Fission is at a first approximation not expected to augment the blue area.*

We appreciate the need to show more evidence for droplet fission linked to the formation of the templated surfactant. We have included plots of the number of droplets, and the mean area per droplet against reaction time as Supplementary Figures 8 and 9 respectively. The number of droplets increases with reaction time, though the deviation between repeats is rather high due to the randomness of droplet division. The mean droplet area increases rapidly at the start as more droplets can “survive” after being placed, then decreases as these droplets undergo fission. This is consistent with the images of droplets at the top of Figure 5.

We have also added explanations to the figure legend and the main text to refer to these pieces of data, and hope that this better conveys our reasoning.

18. The robotics employed is rather trivial and its description should be in the Methods and the figure – in the supporting material.

In response to comment 16), we replaced figure 4 with a flow diagram, and since this provides an overall description of the experiments carried out with the robot, we have moved the specific details about its construction to the Methods section.

19. In the text under legend to Fig. 4, the discussion on equilibrium and away from equilibrium is

rather obscure, may be inaccurate, and is not sufficiently rigorous to become relevant to the main points of the paper.

The idea behind this discussion was to highlight that without the robot, the equilibrium of the imine condensation in contact with the aqueous phase would be pushed towards the reactants by the large amounts of water, and the robot allows the effect of the products to be studied before they are hydrolysed. We agree that this is not central to the main points of the paper, and in fact caused confusion with the previous discussion of the equilibrium concentration of surfactant at the chloroform/water interface, so this discussion was simplified to:

“The robot conducts protocell functions that cannot be fulfilled by the chemistry, in this case protecting the bulk reaction mixture from hydrolysis.”

20. In the conclusions, the author relate to freeing the experiment system from robotics. I assume this means that the templating reaction, now in a separate vessel and dropped in by robot, will become an integral part of what happens in the compartment. This should be more clearly stated, and allusion should be made to the exciting possibility that the templated amphiphile will in itself form the compartment (e.g. in the form of a bilayer).

We are pleased to be able elaborate on this possibility as it represents the next steps for this type of chemistry following on from our proof of concept. We have added the following to the end of the conclusion:

“Specifically, a water stable amphiphilic replicator that could form within a droplet would allow the replication reaction to affect the compartment in real time. An even more advanced system where the reaction could take place in aqueous solution could potentially remove the need for an organic phase, with the replicator itself forming bilayer vesicles. These improvements represent worthwhile next steps in the experimental study of linked template self-replication and compartmentalization.”

Reviewer #3 (Remarks to the Author):

The authors describe an attempt for fulfilling one of the requirements for a chemical system to be a “minimal cellular system” according to the chemoton theory of Tibor Ganti. In this theory, a minimum model of a chemoton has three self-producing systems which are coupled together stoichiometrically: metabolism cycle, template polycondensation, and membrane formation. On paper, the chemoton theory is plausible. However, an experimental realization of this theoretical view is a big challenge, and if one would succeed, one could claim that a living system (life) could be made from bottom-up. This has not been realized yet. The authors carried out experiments for coupling template copying with compartment multiplication for developing a droplet-based model of a protocell. More experimental work in this field should be carried out, independent on whether the molecules and conditions used are prebiotic plausible or not, so that new ideas concerning the origin of the first cells, i.e. the origin of life, hopefully emerge. In this sense, the work presented is a valuable contribution. I recommend considering acceptance of the manuscript once the following points are taken into account (major revision).

1. General: It requires some time to fully understand the details of the system investigated. Therefore, things should be made clearer at different places, including in the abstract. It should be mentioned that the system described is an oil-in-water droplet system and that hydrophobic molecules which are not (or only weakly) amphiphilic are dissolved inside the droplets are converted into amphiphilic molecules via a reaction with a water soluble molecule which takes place through a template mechanism inside the droplets. Is this what the system is?

The reviewer's understanding of the system we investigated is correct. We have modified the abstract to specify that this is an oil-in-water droplet system, and our other revisions to the manuscript, particularly Figure 1 should hopefully make this more intuitive to the reader.

2. Figure 1: For a, the solvent should be mentioned. For b and c, it should be made clear that the droplets shown contain chloroform inside (okay?). What is the typical size of the droplet? For b it is not clear what the very small dots represent.

We have modified Figure 1 according to this comment, and to those of reviewer 2. Particularly, the three panels now depict each of the three studies we carried out on the system (kinetics, DLS, and droplet placement).

For panel a), since it depicts the general idea of an amphiphilic molecule being formed via a template self-replication reaction, it would not make sense to us to specify the solvent. In Figure 2, where the reaction scheme with the chemical structures we used in this study is shown, we now specify that the reaction occurs in chloroform.

For b) – this is meant to show the general idea of reverse micelle formation using an amphiphilic replicator, and the details of the experiments we carried out including solvent system and size of reverse micelles we obtained is best left to the DLS section in our opinion, since different amphiphilic replicators could be possible that operate in different solvents and form differently sized reverse micelles. We have specified in the legend that we demonstrated this in chloroform though.

For c) – similarly, the general concept of observing oil-in-water droplet fission can be investigated on a range of different scales – some may be observed with microscopic droplets, others with droplets of 100 μl in volume. It is now specified in the legend however that ours were 4 μl in volume when placed.

3. Title: ... model protocell ... It is a chemical model system of a protocell. Nobody knows how such (hypothetical) protocells looked like.

We agree with the reviewer that this system is a model system of a protocell, and have changed the title to:

“Autonomous model protocell division driven by molecular replication”

4. *Figure 2, legend. It is not clear whether the entire Figure is taken from ref. 12, or only the templating part. It seems that the work presented is an extension of the work presented in ref. 12. If so, this should be clearly mentioned. Please clarify! It is conceptually important that 1 is not (or only weakly) amphiphilic, while 3 is amphiphilic.*

We have clarified in the figure legend that the figure and compounds **1** and **3** are of our own design, but that the existing replication motif is extended upon from ref. 12. We also agree that it is part of the core concept of the paper that **3** is amphiphilic and **1** is only weakly amphiphilic, and have explained this in the figure legend.

5. *Text on p. 5, above Figure 3 is not clear. Is a word missing? ... the reaction could take place in a dispersed organic phase ...*

We agree with the referee's suggestion, and have modified this to "the reaction could take place in a dispersed organic phase"

6. *Figure 3: The solvents used should be mentioned.*

We have added that the kinetics were investigated in CDCl_3 .

7. *Dynamic light scattering: It should be better argued and clearly explained why reverse micelles were used, why not an oil in water systems, a water-in-oil microemulsion.*

We used such a small volume fraction of water to limit hydrolysis of the imine. Ideally we would have kept the solution dry, but some water is required to form the core of reverse micelles. This has been added to the text.

8. *Page 8: a hydrophobic dye (sudan blue II) ...*

We agree that it makes sense to specify that this is a hydrophobic dye, and have changed this in the text.

9. *Figure 4a: The blue solution should be explained: reaction systems composed of chloroform droplets in water (okay?).*

Figure 4b: Video observation, the same as observation via webcam in Figure 4c?

The drawing in Figure 4c is not clear. Why a blue background? Moveable white background?

What is the orange part.

It is not seen in Figure 4b?

In response to this comment, and those of reviewer 2, Figure 4 was redesigned to give better explanation of the experimental method for the periodic droplet experiments. The blue solution is labelled, and the orange part was removed since this was essentially a rack for vials and the petri dish, and not directly relevant to the experimental design.

10. Methods: All chemicals purification. The sentence is written two times. Copy paste in the Supporting Information, also two times.

We apologize for the error, and have corrected this in the manuscript and the SI.

REVIEWERS' COMMENTS:

Reviewer #1 (Remarks to the Author):

I have agreed with the author's replies (from 1 to 5), except the last one.

I am still concerned the meaning of division of oil droplets containing surfactant 3.

The author should clearly define a protocell in this manuscript. As far as I understood, this oil droplet is designated by authors as a protocell. First, please specify the content of oil droplet. It's a chloroform solution of hydrophobic amine 1, 4-formyl(phenyl)acetic acid 2, and a hydrophobic fluorescence dye, the surface of the droplet being covered with amphiphilic imine 3?

As far as I understood, the droplet spontaneously divides into pieces (compartmentalization) when a chloroform solution was dropped to a water phase. The size of divided droplets depends on the amount of surfactant 3 which is synthesized from 1 and 2 using 3 as a template (molecular replication). Besides, the number of divided droplets increases and the size of the droplets decreases as the amount of 3 in the droplet increases during the interval. I wonder this behavior can be designate as a self-replication? The size of the droplet becomes smaller and smaller as the concentration of 3 increases, and this phenomenon can be expressed in terms of languages of materials science.

The concentration dependence of the size of oil droplets on the amount of the added surfactants is well established. For example, this phenomenon is precisely discussed in the following literature. Slavka Tcholakova, Nikolai D. Denkov, and Thomas Danner, Role of Surfactant Type and Concentration for the Mean Drop Size during Emulsification in Turbulent Flow, *Langmuir* 2004, 20, 7444-7458.

This phenomenon was recently utilized for preparation of phototactic oil droplets of a suitable size. Kentaro Suzuki, Tadashi Sugawara, Phototaxis of Oil Droplets Comprising a Caged Fatty Acid Tightly Linked to Internal Convection, *ChemPhysChem*, 17, 2300 – 2303 (2016); Corresponding data, in particular, are summarized in Figure S1 in Supporting Information.

If some explanation or comments on this point are given in the text, I am happy to accept this manuscript in Nature communication.

Reviewer #3 (Remarks to the Author):

The authors made a substantial revision of the manuscript following the suggestions made by the three reviewers. With this improvement, the manuscript is now clearer and I recommend acceptance after only a few minor points have been considered.

1. Figure 1. The drawing and the legend are very general for illustrating the system investigated. For this reason, I recommend to replace the specific indication of "... 4 uL droplets ..." by, for example, "...small volume droplets ...". Details about the droplet volumes can be given in the legend of Fig. 4.

2. p-tolualdehyde: p should be in italic; n-dodecyl: n should be in italic

3. The precision of the values given for the association constants, 80457 M⁻¹ vs. 34.47 M⁻¹, does not make sense if there is no standard deviation. Please only provide values with an appropriate number of significant digits!

4. Figure 4 should contain more information so that it is immediately clear. Suggestion: label the syringe, label the Petri dish. Does the Petri dish contain some solution? If yes, indicate which solution at which concentration and which volume. Add the values for the droplet volumes. The schematic of the dish cleaning device is not clear. What should it be, a pipette? What does the orange part mean?

5. Figure 5: The scale bar is difficult to see. White may be better than black.

6. Last line before the conclusions section. What do these values mean? F, df, p?

Referee comments in italics; our reply in normal type

Reviewer #1 (Remarks to the Author):

I have agreed with the author's replies (from 1 to 5), except the last one.

I am still concerned the meaning of division of oil droplets containing surfactant 3.

The author should clearly define a protocell in this manuscript. As far as I understood, this oil droplet is designated by authors as a protocell.

We have added a definition of a protocell as follows:

"Such "model protocells" would allow the idea that life on earth emerged from a minimal self-organized cell-like entity – a protocell, to be tested experimentally."

First, please specify the content of oil droplet. It's a chloroform solution of hydrophobic amine 1, 4-formyl(phenyl)acetic acid 2, and a hydrophobic fluorescence dye, the surface of the droplet being covered with amphiphilic imine 3?

We have now specified that this is an oil-in-water droplet at the end of the introduction, and also specified the contents of the droplet again in the legend of Figure 4 in addition to the text of the Results section.

"Droplets of this chloroform solution of reactants 1 and 2, amphiphilic product 3, and the hydrophobic dye are placed into the alkaline aqueous phase and observed from below."

As far as I understood, the droplet spontaneously divides into pieces (compartmentalization) when a chloroform solution was dropped to a water phase. The size of divided droplets depends on the amount of surfactant 3 which is synthesized from 1 and 2 using 3 as a template (molecular replication). Besides, the number of divided droplets increases and the size of the droplets decreases as the amount of 3 in the droplet increases during the interval. I wonder this behavior can be designate as a self-replication? The size of the droplet becomes smaller and smaller as the concentration of 3 increases, and this phenomenon can be expressed in terms of languages of materials science.

The concentration dependence of the size of oil droplets on the amount of the added surfactants is well established. For example, this phenomenon is precisely discussed in the following literature. Slavka Tcholakova, Nikolai D. Denkov, and Thomas Danner, Role of Surfactant Type and Concentration for the Mean Drop Size during Emulsification in Turbulent Flow, Langmuir 2004, 20, 7444-7458.

This phenomenon was recently utilized for preparation of phototactic oil droplets of a suitable size. Kentaro Suzuki, Tadashi Sugawara, Phototaxis of Oil Droplets Comprising a Caged Fatty Acid Tightly Linked to Internal Convection, ChemPhysChem, 17, 2300 – 2303 (2016); Corresponding data, in particular, are summarized in Figure S1 in Supporting Information.

These references are useful in explaining why the surfactant concentration is coupled to droplet division, and we have added the most recent one to the discussion to explain the physical basis behind what we observe.

We do not yet refer to the droplet division as self-replication, but we demonstrate that a template self-replication reaction can drive droplet division through this mechanism, and in principle along with a mechanism for droplet growth by assimilating material from its surroundings, this would lead to a true droplet self-replication cycle coupled to molecular template self-replication. This is the current direction of our further research.

If some explanation or comments on this point are given in the text, I am happy to accept this manuscript in Nature communication.

We have added the following to the last paragraph of the discussion, including the most recent suggested reference:

“The relationship between surfactant concentration and droplet size for micron scale droplets has been well investigated previously, and droplet size is known to be highly sensitive to even low concentrations of surfactant.²⁰ The application of this principle to millimetre scale droplets, and its use to allow template self-replication to drive droplet division, forms a protocell model where template self-replication is linked to compartmentalization...”

Reviewer #3 (Remarks to the Author):

The authors made a substantial revision of the manuscript following the suggestions made by the three reviewers. With this improvement, the manuscript is now clearer and I recommend acceptance after only a few minor points have been considered.

1. Figure 1. The drawing and the legend are very general for illustrating the system investigated. For this reason, I recommend to replace the specific indication of “... 4 uL droplets ...” by, for example, “...small volume droplets”. Details about the droplet volumes can be given in the legend of Fig. 4.

We have changed “4 uL droplets” to “microliter volume droplets” to give an idea of scale, but still remain general.

2. p-tolualdehyde: p should be in italic; n-dodecyl: n should be in italic

We have corrected this in the manuscript and SI.

3. The precision of the values given for the association constants, 80457 M⁻¹ vs. 34.47 M⁻¹, does not

make sense if there is no standard deviation. Please only provide values with an appropriate number of significant digits!

We have changed to using 2 *sf* for the association constants to reflect the precision of the rate constants they were calculated from.

4. Figure 4 should contain more information so that it is immediately clear. Suggestion: label the syringe, label the Petri dish. Does the Petri dish contain some solution? If yes, indicate which solution at which concentration and which volume. Add the values for the droplet volumes. The schematic of the dish cleaning device is not clear. What should it be, a pipette? What does the orange part mean?

We have added additional labels to Figure 4, and changed the drawing of the dish cleaning device to clarify that it consists of two tubes, one to dispense acetone, the other to aspirate the dish contents.

5. Figure 5: The scale bar is difficult to see. White may be better than black.

We have changed the scale bar colour to white.

6. Last line before the conclusions section. What do these values mean? F, df, p?

These are related to the single factor ANOVA performed on the data, we have added a brief explanation:

(ANOVA F factor = 18.6, degrees of freedom = 89, p value = 4.28×10^{-5}).